# Fluorescence Microscopy and Flow-Cytometry Assessment of Substructures in European Red Deer Epididymal Spermatozoa after Cryopreservation

**DOI:** 10.3390/ani13060990

**Published:** 2023-03-08

**Authors:** Anna Dziekońska, Marek Lecewicz, Agnieszka Partyka, Wojciech Niżański

**Affiliations:** 1Department of Animal Biochemistry and Biotechnology, University of Warmia and Mazury in Olsztyn, Oczapowskiego 5, 10-719 Olsztyn, Poland; 2Department of Reproduction and Clinic of Farm Animals, Wroclaw University of Environmental and Life Sciences, Pl. Grunwaldzki 49, 50-366 Wroclaw, Poland

**Keywords:** epididymal sperm, cryopreservation, flow cytometry, microscopic analysis

## Abstract

**Simple Summary:**

Flow cytometry (FC) is the recommended technique for assessing sperm quality. In comparison with fluorescence microscopy (FM), FC supports analyses of much larger sperm populations and generates more reliable results. However, FC is not always accessible, and sperm populations are often evaluated with the use of FM. In the present study, FC and FM were used to assess the functionality of various organelles in European red deer epididymal spermatozoa stored in liquid nitrogen. Spermatozoa were collected from the epididymides of hunter-harvested European red deer stags. The epididymides were stored in a refrigerator (2–4 °C) for 12 h before analysis. The study demonstrated that the refrigerated storage of the epididymides for 12 h had no significant effect on the sperm quality before cryopreservation, but it significantly influenced the percentage of early necrotic sperm after thawing. The results of FM and FC assays differed significantly, excluding in the assessment of the plasma membrane integrity. However, the results of both assays revealed significant correlations between the examined variables, except for mitochondrial activity. The study demonstrated that the spermatozoa from epididymides chill-stored for 12 h can be used for cryopreservation. Fluorescence microscopy and FC are equally reliable techniques, but FM was more useful for evaluating mitochondrial activity.

**Abstract:**

Thawed spermatozoa, sampled post mortem from the fresh epididymides of European red deer and epididymides stored for up to 12 h at 2–4 °C, were evaluated by fluorescence microscopy (FM) and flow cytometry (FC). The sperm samples were extended and cryopreserved. The sperm motility (CASA), sperm viability (SYBR^+^/PI^-^), acrosome integrity, mitochondrial activity, apoptotic changes, and chromatin stability were assessed. Sperm were analyzed by FM before cryopreservation, and by FM and FC after thawing. Epididymal storage time (for 12 h) had no significant effect (*p* > 0.05) on the examined variables before cryopreservation. After thawing, the storage variants differed (*p* ˂ 0.05) in the percentage of apoptotic sperm (FM and FC) and DNA integrity (FC). The results of FM and FC differed (*p* ˂ 0.05) in all the analyzed parameters, excluding SYBR^+^/PI. Significant correlations (*p* ˂ 0.01) were observed between the sperm viability, acrosome integrity, and the percentage of non-apoptotic spermatozoa, regardless of the applied technique. In FM, the above parameters were also significantly correlated with mitochondrial activity. The study demonstrated that European red deer spermatozoa stored in the epididymides at 2–4 °C for 12 h can be used for cryopreservation. Both techniques were equally reliable, but FM was better suited for evaluating mitochondrial activity whereas FC was more useful in the evaluation of DNA fragmentation.

## 1. Introduction

Sperm collected from hunter-harvested wild-living animals is a valuable source of genetic material that can be used for the conservation and genetic improvement of animal populations in breeding farms [1,2,3]. Sperm can be cryopreserved in liquid nitrogen to preserve its fertilizing potential for a long period of time (many years), which has important implications for species conservation [4,5,6].

Ejaculated sperm is the most recommended for reproductive purposes, but it is not easy to acquire, and epididymal sperm can also be used [7]. In wild-living animals, post-mortem sperm sampling from the epididymides is the easiest sperm-harvesting method, which, unlike electroejaculation, does not require specialist equipment or complex procedures [8,9]. The epididymal sperm for reproductive purposes can be stored in a liquid state or cryopreserved [4,10,11]. However, epididymal spermatozoa cannot always be collected directly from hunter-harvested animals because the hunting grounds are usually situated in remote locations. As a result, the sperm are harvested several hours after the hunt. The collected material is transported to a laboratory and stored in a refrigerator at a temperature of 0–6 °C and 80–85% humidity. The storage conditions can significantly affect the viability of the epididymal sperm, and further research is needed to address this aspect.

Previous studies have shown that the sperm of many animal species, including mice [12], rams [13], dogs [14], and bulls [15,16,17], is resistant to low storage temperatures in the range of 0–6 °C. Iberian red deer spermatozoa can be stored in the epididymides for up to several days at a temperature 4–5 °C [7,18] before cryopreservation [19,20]. These observations suggest that European red deer spermatozoa intended for cryopreservation should also tolerate temperatures in the range of 2–4 °C.

Sperm quality is evaluated with the use of various laboratory methods, where fluorescence techniques, motility analyses, and morphology assessments are most recommended. These assays are conducted with the use of a fluorescence microscope (FM) or a flow cytometer (FC). Each technique has its strengths and weaknesses [21]. Fluorescence microscopy is a time-consuming method that should be performed by a qualified and experienced observer, and it involves time limits and a small number of cells [22]. In turn, FC supports accurate and simultaneous assessments of various cell structures [23,24]. Flow cytometry is highly recommended in clinical trials because it supports rapid and objective analyses of large cell samples [22,25]. Flow cytometry is also recommended for fluorescence analyses of sperm quality [23]. However, flow cytometers are expensive and not readily accessible. Most laboratories are equipped with fluorescence microscopes, but only small sperm samples (200–300 cells) are analyzed under an FM. According to many researchers, these limitations undermine the reliability of the results.

In view of the above, the aim of this study was to evaluate different structures in cryopreserved epididymal spermatozoa of European red deer, collected directly from fresh epididymides and from epididymides stored for 12 h at a temperature of 2–4 °C, with the use of FM and FC. Sperm motility and motility parameters were determined, and fluorescence analyses of sperm viability, acrosomal membrane integrity, mitochondrial membrane potential, apoptotic changes, and DNA status were performed. Spermatozoa were subjected to FM only before cryopreservation, and to both FM and FC after thawing. The same fluorochromes were used in FM and FC.

## 2. Materials and Methods

### 2.1. Animals

Red deer stags were hunted in accordance with the harvest plan for each species of game animals, in the in the Forest District of Nowe Ramuki (Region of Warmia and Mazury, Poland), in September and October (during the rutting season), in accordance with the Polish Hunting Law. Experimental material was collected post mortem, and it consisted of the testes and epididymides that were stored in the scrotum until sperm collection. The sperm samples were collected from the epididymides of 16 European red deer stags (aged 4 to 11 years, with body weight of 140 kg to approx. 180 kg).

### 2.2. Sampling and Cryopreservation of Spermatozoa

Spermatozoa were collected from the tail of the epididymis (within 2 h post mortem, variant I) and from epididymides refrigerated for 12 h (2–4 °C, variant II). Spermatozoa were sampled according to a previously described procedure [26]. In the collected samples, the sperm motility was assessed using the computer-assisted semen analysis (CASA) system and the sperm concentration was evaluated using a Bürker counting chamber (Equimed-Medical Instruments, Cracow, Poland).

The spermatozoa were cryopreserved according to the procedure described by Martínez-Pastor et al. [27], with some modifications. In the first trial, the samples were diluted with a commercial freezing extender (Andromed, Minitub GmbH, Tiefenbach, Germany) to a concentration of 400 × 10^6^ cells/mL and transferred to a refrigerator (4 °C) for 2 h to equalize the temperature. Then, the samples were diluted with the same extender to a concentration of 160–200 × 10^6^ cells/mL and left to stand for 1 h. The samples were then packed in 0.25 mL straws (IMV, L’Aigle Cedex, France) and frozen in nitrogen vapor (4 cm above liquid nitrogen) for 10 min. The straws were transferred to liquid nitrogen, where they were kept for at least 1 year. The samples were thawed by placing the straws in water (65 °C) for 6 s.

### 2.3. Sperm Analysis

#### 2.3.1. Sperm Motility

Sperm motility was evaluated using the CASA system (Hamilton Thorne Sperm Analyzer IVOS version 12.2l; Hamilton Thorne Biosciences, MA, USA). Sperm samples were diluted 1 : 100 (fresh) and 1 : 5 (fresh diluted sperm, frozen-thawed) in phosphate-buffered saline (PBS) to a concentration of around 30–50 × 10^6^ cells/mL. A 5 μL aliquot of the sample was placed in a pre-warmed Makler counting chamber (Sefi-Medical Instruments Ltd., Haifa, Israel) and evaluated at 37 °C. In each sample, spermatozoa were analyzed in five fields of view selected randomly by the computer. The analyses included the determination of: percentage of total motile sperm (TMOT), percentage of progressive motile sperm (PMOT), and parameters characterizing sperm movement: VAP, VSL, VCL, ALH, BCF, LIN (VSL/VCL ratio × 100 %), and straightness (STR, VSL/VAP ratio × 100 %).

The CASA system settings recommended by Hamilton Thorne for gazelle/deer [26] were used in the analyses. Progressive motility was defined as the percentage of spermatozoa with a VAP > 75.0 μm/s and an STR > 80%.

#### 2.3.2. Fluorescence Assay

The sperm characteristics were evaluated by the fluorescence method before and after cryopreservation. The Olympus BX41 Fluorescence Microscope, which offers ultraviolet (330–385 nm), blue (460–490 nm), and green (510–550 nm) excitation wavelengths, was used for fluorescence analyses. Stained samples were analyzed at 600 × magnification. A minimum of 300 cells per slide were examined in each aliquot.

The viability (plasma membrane integrity) was assessed using SYBR-14 and propidium iodide (PI) fluorescent probes (Live/Dead Sperm Viability Kit; Life Technologies Ltd., Grand Island, NY, USA) according to a previously described method [26]. The diluted sperm samples (200 μL) were supplemented with 2 μL of 1 mM SYBR-14 solution and 2 μL of PI (2.4 μM in Tyrode’s salt solution), and incubated at 37 °C in the dark for 10 min. Sperm with green heads (SYBR-14^+^/PI^−^, live sperm with integral membranes) and sperm with red heads (dead sperm) were identified.

The acrosomal status of spermatozoa was evaluated using fluorescein isothiocyanate-labeled peanut (*Arachis hypogaea*) agglutinin (FITC-PNA; Life Technologies Ltd., Grand Island, NY, USA) with a PI solution. The method was described in earlier reports [11,28]. Next, 1 µL of JC-1 (1 mg JC-1/mL anhydrous dimethyl sulfoxide, DMSO) was added to diluted sperm samples (200 μL), and they were incubated at 37 °C for 15 min (to identify all sperm cells under the fluorescence microscope). Then, 2 μL of FITC-PNA solution and 2 μL of PI were added to the samples, which were incubated again for 5 min at 37 °C. The FITC-PNA working solution was prepared by dissolving 2 mg of FITC-PNA in 1 mL of PBS. Four sperm populations were observed under the microscope: non-stained spermatozoa in the head region with fluorescence in the midpiece were classified as live sperm with intact acrosomes (FITC-PNA^−^/PI^−^); sperm with green-red fluorescence in the head (FITC-PNA^+^/PI^+^) were classified as early necrotic, acrosome-reacted sperm; sperm with a red head were considered as dead; and a small population of spermatozoa had only green staining of the acrosomal cap (acrosome-reacted sperm) (Figure 1A).

The mitochondrial activity was evaluated by examining the mitochondrial membrane potential (MMP) of sperm with the use of JC-1 fluorochromes (Life Technologies Ltd., Grand Island, NY, USA) with PI, according to a previously described method [11,28]. Diluted sperm samples (200 μL) were incubated with 1 μL of JC-1 solution for 15 min at 37 °C. Next, 1 μL of PI was added to the samples, which were incubated again for 5 min at 37 °C. The analysis revealed the presence of sperm with orange midpieces (with active mitochondria, high MMP) and sperm with green midpieces or unstained, often with red fluorescence in the head region (low MMP) (Figure 1B).

The DNA integrity was assessed using acridine orange, as previously described by Partyka et al. [29] with some modifications. Sperm samples (50 μL) were subjected to acid denaturation (30 s) by adding 200 μL of a lysis buffer (Triton X-100 0.1% (*v*/*v*), NaCl 0.15 M, HCl 0.08 M, pH 1.4). Subsequently, 600 µL of acridine orange solution was added to the samples, which were incubated for 3 min in the dark. The fluorescence analysis revealed the presence of sperm with green heads (with integral DNA) and sperm with orange or red heads (with DNA fragmentation; DFI) (Figure 1C,D).

The Vybrant Apoptosis Assay Kit #4 (Life Technologies Ltd., Grand Island, NY, USA) was used to assess apoptosis and membrane integrity according to a previously described method [11,28]. First, to visualize all cells under a fluorescence microscope, the diluted samples (200 µL) were stained with 1 µL JC-1 (as described above). Then, 2 μL of YO-PRO-1 and 2 μL of PI were added to the samples, which were incubated at 37 °C for 5 min. Four sperm populations were distinguished in the fluorescence analysis: viable sperm (YO-PRO-1^−^/PI^−^), sperm showing apoptotic-like changes (YO-PRO-1^+^/PI^−^), sperm with early necrotic changes (YO-PRO-1^+^/PI^+^), and dead/necrotic sperm (YO-PRO-1^−^/PI^+^), according to a previously described staining model [11].

#### 2.3.3. Flow-Cytometry Analysis

Flow-cytometry analyses were performed in a Guava EasyCyte 5 (Merck KGaA, Darmstadt, Germany) cytometer. Fluorescence was induced by an argon ion 488 nm laser. Data were acquired using the GuavaSoft™ 3.1.1 software (Merck KGaA, Darmstadt, Germany). Non-sperm events were gated out based on scatter properties and excluded from analyses. A total of 10,000 events were analyzed for each sample. For the flow-cytometry analysis, the stained sperm samples were extended to a concentration of around 5–10 × 10^6^ spermatozoa/1 mL. The same fluorochromes were used in fluorescence flow-cytometry analyses and in the microscopic analysis.

To assess the plasma membrane integrity, spermatozoa were stained with SYBR-14 and propidium iodide (PI) according to the protocol described by Partyka et al. [29]. The diluted samples (500 μL) were stained with 3 μL of SYBR-14 and 3 μL of PI. SYBR-14-positive sperm with green fluorescence and PI-negative sperm were classified as live sperm cells with intact plasma membranes.

Sperm acrosome status was assessed with lectin PNA from *Arachis hypogaea* conjugated with Alexa Fluor^®^ 488 (Life Technologies Ltd., Grand Island, NY, USA). The diluted semen samples were combined with 10 μL of PNA working solution (1 μg/mL) and incubated for 5 min at room temperature in the dark. After incubation, the samples were washed, and 3 μL of PI was added before the fluorescence analysis [29].

Sperm mitochondrial activity was determined by JC-1 and PI staining. A 3 mM stock solution of JC-1 in DMSO was prepared. Diluted sperm samples (500 μL) with a concentration of 50 × 10^6^ cells/mL were combined with 0.67 μL of JC-1. Before analysis, the probes were incubated for 20 min, in the dark, at 37 °C [30]. Sperm emitting orange fluorescence were categorized as cells with high mitochondrial potential (HMP), whereas sperm emitting green fluorescence were classified as cells with low mitochondrial potential (LMP).

Apoptosis and sperm viability were evaluated with YO-PRO-1 (25 μM solution in DMSO) and PI fluorochromes. Sperm were diluted with PBS to a concentration of 50 × 10^6^ cells/mL. A diluted sample of 1 mL was stained with 1 μL of YO-PRO-1 (with a final concentration of 25 nM) and 1 μL of PI [31]. In the FC analysis, fluorescence was measured with a FL-2 sensor and a 575 nm band-pass filter to detect PI, and with a FL-1 sensor and a 525 nm band-pass filter to detect YO-PRO-1.

Acridine orange was used to assess DNA integrity, as described above, with minor changes. The sperm suspension (100 μL) was denatured with a lysing solution (200 μL) for 30 s, and then AO solution (600 μL) was added. The samples were analyzed after 3 min of incubation. Spermatozoa with normal double-stranded DNA displayed green fluorescence and were regarded as the main population. Spermatozoa with increased red fluorescence indicative of DNA fragmentation (DFI) were located to the right of the main population.

### 2.4. Statistical Analysis

The results were processed with the use of a general linear model (GLM) and ANOVA in the Statistica program (v. 13.3, StatSoft Incorporation, Tulsa, OK, USA). The assumption of normality was checked using the Shapiro–Wilk test, and the homogeneity of variance was assessed with Levene’s test. The data that were not normally distributed were transformed before statistical analysis. The percent data were arcsine transformed. The results were presented as means ± standard error of the mean (SEM). Tukey’s post-hoc test was used to compare the means, and the statistical significance was set at *p* < 0.05. The relationships between sperm motility and fluorescence parameters were examined by calculating the Pearson correlation coefficient.

## 3. Results

The sperm motility analysis revealed that the compared storage variants had no significant effect (*p* > 0.05) on sperm motility parameters both before cryopreservation and after thawing (Table 1). In both variants, cryopreservation induced a significant (*p* ˂ 0.05) decrease in the values of TMOT, PMOT, VAP, VSL, VCL, and ALH. No significant changes were found in the values of BCF, STR, and LIN before and after thawing.

The FM assay revealed that cryopreservation significantly influenced all examined sperm structures, excluding DNA status, in both storage variants (Figure 2 and Figure 3). The variant where sperm were stored in the epididymides for up to 12 h before cryopreservation did not induce significant changes in sperm cell structures, relative to the variant where sperm were sampled directly from fresh epididymides and cryopreserved.

A similar relationship was observed in thawed sperm, excluding the percentage of normal spermatozoa without apoptotic changes (FC), the percentage of apoptotic-like spermatozoa (FM and FC) (Figure 3A,B), and DNA fragmentation (DFI, FC) (Figure 3E,F).

In the thawed sperm, the percentage of spermatozoa with intact plasma membranes was the only parameter that was not significantly (*p* > 0.05) affected by the applied fluorescence technique. The results of FM and FC assays differed significantly (*p* ˂ 0.05) in the following parameters: the percentage of spermatozoa with high mitochondrial membrane potential (Figure 2C,D), acrosome integrity (Figure 2E,F), apoptotic changes (Figure 3A–C), and chromatin stability (Figure 3E,F). In the analysis of mitochondrial activity, the percentage of spermatozoa with a high mitochondrial membrane potential was significantly higher in FM than in FC, regardless of the storage variant (Figure 2C,D). The percentage of sperm with intact acrosomes was significantly higher in FC than in FM (Figure 2E,F). The percentage of non-apoptotic sperm was also significantly higher in FC than in FM (Figure 3A). However, the percentage of apoptotic-like and early necrotic spermatozoa was higher in FC than in FM, irrespective of the storage variant (Figure 3B,C). In the DNA status analysis, a significantly (*p* ˂ 0.05) smaller percentage of spermatozoa with fragmented DNA was identified in FC than in FM (Figure 3E,F).

The correlations between sperm motility and the analyzed fluorescence parameters in FM and FC are presented in Table 2 and Table 3, respectively.

The correlation analysis for FM revealed significant (*p* ˂ 0.05) correlations between the percentage of motile sperm (TMOT) versus acrosomal membrane integrity, the percentage of non-apoptotic sperm, and the percentage of sperm with high mitochondrial membrane potential (HMP). The parameter HMP was also correlated (*p* ˂ 0.01) with plasma membrane integrity, acrosome integrity, and the percentage of non-apoptotic sperm. Progressive motility was significantly (*p* ˂ 0.05) correlated with plasma membrane integrity.

The FC assay also revealed a positive correlation (*p* ˂ 0.05) between TMOT versus acrosome integrity, and between PMOT versus plasma membrane integrity and the percentage of non-apoptotic spermatozoa. Plasma membrane integrity was highly positively correlated (*p* ˂ 0.001) with acrosome integrity, and positively correlated (*p* ˂ 0.01) with the percentage of non-apoptotic sperm. In turn, the HMP was not significantly (*p* > 0.05) correlated with the remaining parameters in the FC assay. No significant correlations were found between the DNA status and the remaining parameters.

## 4. Discussion

The study demonstrated that the storage of European red deer spermatozoa in the epididymides at a temperature of 2–4 °C for up to 12 h did not significantly decrease sperm viability or functionality in comparison with the variant where spermatozoa were sampled directly from fresh epididymides. Sperm motility, motility parameters, plasma membrane integrity, mitochondrial activity, and apoptotic changes were similar in both storage variants. These results confirm the previous observations made in Iberian red deer, and indicate that red deer spermatozoa are relatively resistant to cold shock [7,18,19]. Interestingly, even small fluctuations in temperature in the range of 2–4 °C did not lead to a significant decrease in the quality of spermatozoa stored in the epididymides for 12 h. Possibly, the storage time was too short to induce significant changes in the sperm function. In other studies, significant changes in the motility and structure of spermatozoa stored in the epididymides at a temperature of 4–5 °C were reported only after 24 h [32] or even 48 h [19,20]. However, in the present study, the epididymides were stored in the scrotum, whereas in the cited studies, the epididymides and the testicles were removed from the scrotum, placed in plastic bags, and stored in beakers filled with water at a temperature of 4–5 °C. The above factor could have played a role in preserving high sperm quality before cryopreservation.

Despite considerable advances in freezing and thawing methods, cryopreservation always leads to damage in cell structures, which alters sperm motility, motility parameters, and decreases semen fertilizing ability [33,34]. In the current study, significant changes in all the evaluated parameters were also observed, including the plasma membrane integrity, acrosome integrity, and mitochondrial membrane potential, which decreased the TMOT and, in particular, PMOT values that are influenced by the VAP and VSL values.

After thawing, most of the analyzed variables did not differ significantly between storage variants, which could be attributed to similar sperm quality before cryopreservation. The quality of sperm cells before cryopreservation significantly influences their resistance to freezing damage [19]. The significant differences between the tested storage variants were observed only in the percentage of apoptotic spermatozoa (FM and FC) and DNA fragmentation (DFI, FC). This observation suggests that sperm stored in the epididymides at a temperature of 2–4 °C were more susceptible to oxidative damage and apoptotic changes during cryopreservation. In another study, cryopreservation induced apoptosis in bovine spermatozoa [35]. In boar spermatozoa, freezing and thawing stimulated an excessive generation of reactive oxygen species (ROS), exerted a negative effect on the cell membrane, mitochondrial activity, and chromatin stability, and accelerated apoptosis [36].

Fluorescent microscopy and FC are widely used to assess sperm structures and semen quality. In the present study, the applied fluorescence technique significantly affected the values of most variables, which could be attributed to numerous factors, including the number of sperm cells. A total of 10,000 sperm cells were evaluated by FC, but only 300 spermatozoa were assessed by FM. It should also be noted that sperm sampled from the tail of the epididymis can be contaminated with blood cells, epithelial cells, or tissue fragments, which can compromise the differentiation of stained spermatozoa by FC [22]. Forward-scatter (FSC) and side-scatter (SCC) gating strategies can be applied to eliminate these contaminants based on differences in the size and complexity of the debris and spermatozoa, but this approach does not always generate reliable results [23]. Flow cytometry is the most recommended technique for the fluorescence assessment of cells, but FM is also an accurate and reliable method, regardless of the origin of sperm cells (ejaculate, epididymides, or testicles) [22].

The compared fluorescence techniques produced similar results only with regard to the viability of sperm stained with SYBR-14/PI. Similar observations were made by Merkies et al. [37], where sperm viability evaluated with the same combination of fluorochromes was similar in both FC and FM.

The greatest differences in the results of FM and FC were noted in the percentage of sperm with high mitochondrial membrane potential (HMP) and DNA integrity in thawed spermatozoa. JC-1 staining supports the identification of two or three sperm populations with different mitochondrial membrane potential [23]. Sperm with a medium MMP (green-orange fluorescence) were most difficult to identify in both FM and FC, which could have affected the results. In the FM analysis, sperm with a medium MMP were difficult to distinguish from sperm with HMP (orange fluorescence) and were classified as spermatozoa with HMP. In turn, in the FC analysis, these sperm were classified as spermatozoa with low MMP (Figure 2D). These difficulties contributed to significant differences in HMP values between FM and FC, irrespective of the storage variant. According to some authors, JC-1 is the most sensitive and reliable fluorescent probe for assessing MMP in FC [38,39], but Martínez-Pastor et al. [23] observed that this probe can be highly sensitive to the staining conditions and that controls should be established to select flow-cytometer settings.

The percentage of sperm with intact acrosomal membranes and the percentage of non-apoptotic sperm were lower in FM than in FC, probably due to differences in the number of the analyzed sperm cells. These differences can be also attributed to a lower detection sensitivity (depending on the observer’s visual processing skills), a smaller number of analyzed cells, and the risk of fluorescence decay during the readout [22]. This problem was particularly evident in the analysis of the percentage of early necrotic sperm, which was significantly lower in FM than in FC. The FM assessment revealed a higher percentage of dead than early necrotic sperm, which influenced the final percentage of non-apoptotic spermatozoa.

The percentage of sperm with fragmented DNA was higher in FM than FC, which could be attributed to the difficulties in distinguishing between the green (sperm with normal DNA) and orange fluorescence (sperm with fragmented DNA) in FM. The fluorescence decay in the samples analyzed under a microscope could be another reason. When the samples for DNA analyses are exposed to laser light for a long period of time, sperm cells are probably rapidly degraded. They change color from green to orange, which is indicative of DNA fragmentation. The fluorescence excitation can lead to the formation of highly reactive photochemical products (reactive oxygen species (ROS), heat, and DNA damage), which are influenced by many factors, including excitation wavelength, intensity, and exposure time [40,41,42,43]. Flow-cytometry measurements are conducted rapidly (in less than 1 min), and this technique seems to be more appropriate for evaluating DNA fragmentation than FM. Sample preparation, as well as the time and conditions of denaturation and staining, can also significantly affect DNA integrity [23,44]. In other studies, the results of DNA fragmentation analyses also differed considerably between FM and FC [21,45].

Despite the considerable differences in the values of most analyzed variables, the results of FM and FC were subjected to a correlation analysis that revealed similar relationships between the sperm viability, acrosome integrity, the percentage of normal sperm without apoptotic changes, and sperm motility. Similar observations were made in other studies, where spermatozoa were assessed by FM [19,38] and FC [46,47]. Therefore, both techniques can be effectively used to assess sperm cell structures. The results of the correlation analysis indicate that FM was a more reliable method than FC in determining the percentage of sperm with high MMP.

The present study revealed that both fluorescence techniques have certain limitations that should be taken into consideration when assessing the various sperm structures.

## 5. Conclusions

The results of this study indicate that spermatozoa obtained post mortem and stored in the epididymides for up to 12 h (at 2–4 °C) can be used for cryopreservation. Fluorescence microscopy and FC are equally reliable techniques for assessing various sperm structures. However, the FM method proved to be more useful in the assessment of mitochondrial activity, whereas the FC assay was more useful in the evaluation of DNA fragmentation.

## Figures and Tables

**Figure 1 animals-13-00990-f001:**
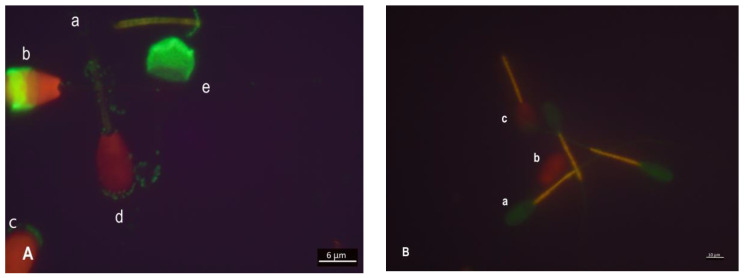
Microscopic evaluation of sperm cell structures using various fluorochromes. (**A**) Acrosome status assessed by FITC-PNA staining: (a) spermatozoa without fluorescence in the head region and with fluorescence in the mid-piece—live sperm with intact acrosomes (FITC-PNA^-^/PI^-^); (b) sperm with green-red fluorescence in the head—early necrotic, acrosome-reacted sperm; (c,d) sperm with red fluorescence in the head—dead sperm; (e) sperm with green fluorescence in the acrosomal cap—acrosome-reacted sperm. (**B**) Mitochondrial membrane potential assessed with the use of JC-1/PI: (a) sperm with green fluorescence in the head and orange fluorescence in the midpiece—live sperm with high mitochondrial membrane potential; (b) sperm with red fluorescence in the head and green fluorescence or no fluorescence in the midpiece—dead sperm with low mitochondrial membrane potential; (c) sperm with green-red fluorescence in the head and orange fluorescence in the midpiece—early necrotic sperm with high mitochondrial membrane potential. (**C**,**D**) DNA status assessed by acridine orange staining: (a) sperm with green fluorescence in the head— normal sperm; (b) sperm with red or orange fluorescence in the head—sperm with damaged DNA (DNA fragmentation, DFI).

**Figure 2 animals-13-00990-f002:**
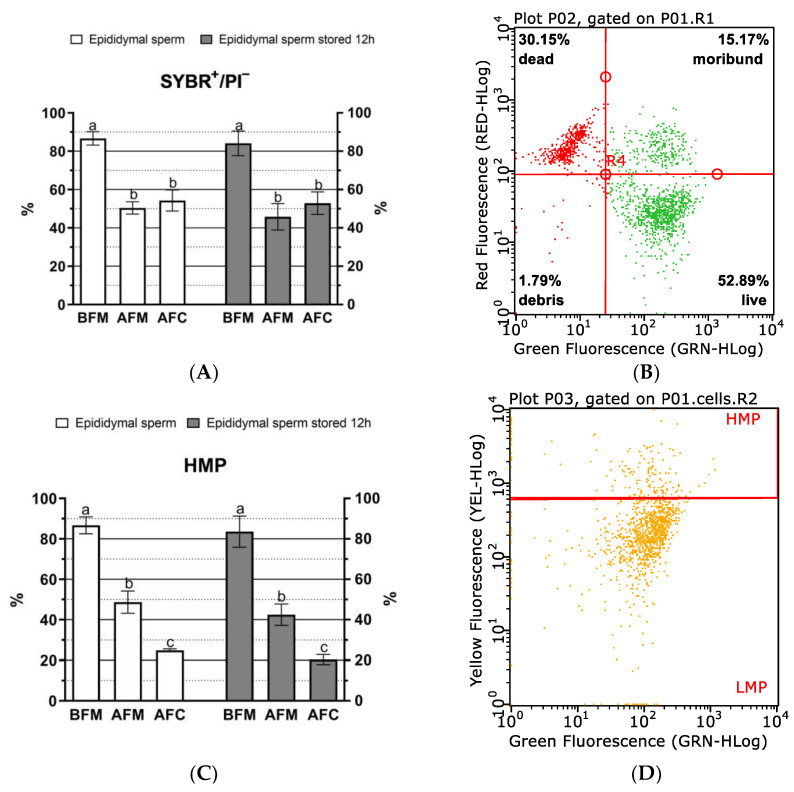
(**A**) Sperm viability (SYBR^+^/PI^−^). (**C**) Percentage of sperm with high mitochondrial membrane potential (HMP). (**E**) Acrosome membrane integrity (PNA^−^/PI^−^). Red deer epididymal spermatozoa were examined under a fluorescence microscope (BFM) before cryopreservation, and under a fluorescence microscope (AFM) and in a flow cytometer (AFC) after thawing. Values represent means (±SEM). Values with different letters (a–c) denote significant differences between treatments at *p* < 0.05 (Tukey’s post-hoc test). (**B**) Cytogram of a SYBR-14/PI stain. (**D**) Cytogram from the analysis of mitochondrial membrane potential of thawed sperm using the JC-1 stain. (**F**) Cytogram from the analysis of acrosome status using the PNA/PI stain.

**Figure 3 animals-13-00990-f003:**
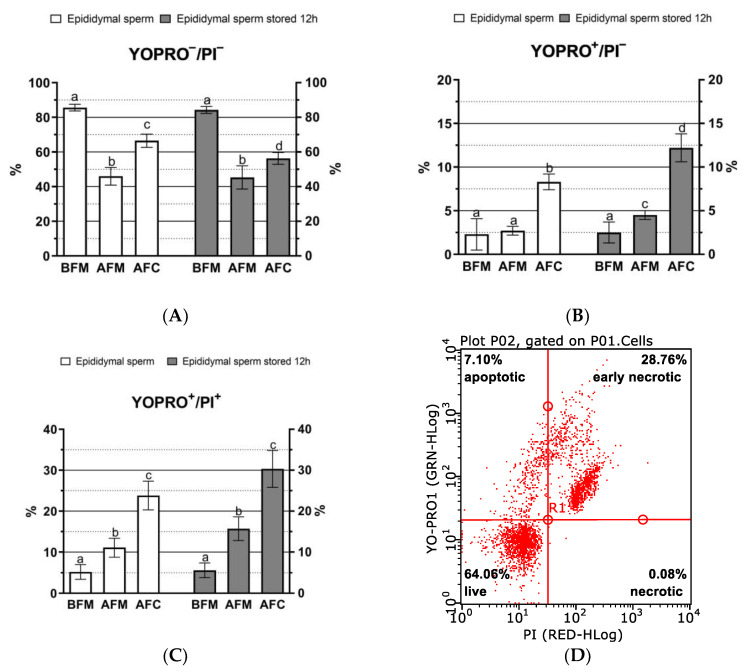
(**A**) Percentage of normal sperm without apoptotic changes. (**B**) Percentage of apoptotic-like sperm. (**C**) Percentage of early necrotic sperm. (**E**) Percentage of sperm with DNA fragmentation. Red deer epididymal spermatozoa were examined under a fluorescence microscope (BFM) before cryopreservation, and under a fluorescence microscope (AFM) and in a flow cytometer (AFC) after thawing. Values represent means (±SEM). Values with different letters (a–c) denote significant differences between treatments at *p* < 0.05 (Tukey post-hoc test). (**D**) Cytogram from the analysis of apoptotic changes in thawed sperm using the YOPRO-1/PI stain. (**F**) Cytogram from the analysis of the DNA status of thawed sperm using the orange acridine stain.

**Table 1 animals-13-00990-t001:** Motility parameters of red deer epididymal spermatozoa (*n* = 8) before and after cryopreservation.

Sperm Motility Parameters	Epididymal Sperm		Epididymal Sperm Stored for 12 h	
	Before Cryopreservation	After Cryopreservation	Before Cryopreservation	After Cryopreservation
TMOT (%)	79.9 ± 1.2 ^a^	51.8 ± 7.5 ^b^	78.5 ± 1.4 ^a^	47.3 ± 6.2 ^b^
PMOT (%)	41.5 ± 0.8 ^a^	24.1 ± 4.5 ^b^	41.9 ± 1.1 ^a^	18.1 ± 5.1 ^b^
VAP (µm/s)	131.5 ± 4.9 ^a^	109.5 ± 7.1 ^b^	126.4 ± 5.2 ^a^	110.3 ± 7.2 ^b^
VSL (µm/s)	85.7 ± 3.2 ^a^	72.2 ± 5.6 ^b^	81.1 ± 2.1 ^a^	72.5 ± 5.6 ^b^
VCL (µm/s)	253.8 ± 9.3 ^a^	215.9 ± 14.7 ^b^	243.1 ± 10.4 ^a^	220.7 ± 15.7 ^b^
ALH (µm)	9.4 ± 0.3 ^a^	8.2 ± 0.3 ^b^	8.9 ± 0.3 ^a^	8.3 ± 0.4 ^a^
BCF (Hz)	33.1 ± 8.7 ^a^	33.6 ± 1.7 ^a^	40.7 ± 5.4 ^a^	33.6 ± 1.7 ^a^
STR (%)	65.4 ± 2.2 ^a^	65.3 ± 2.7 ^ab^	64.8 ± 3.8 ^a^	65.4 ± 2.8 ^a^
LIN (%)	35.1 ± 1.3 ^a^	35.2 ± 2.7 ^a^	35.1 ± 2.3 ^a^	34.9 ± 2.8 ^a^

TMOT, total motility. PMOT, progressive motility. VAP, velocity average path. VSL, velocity straight line. VCL, curvilinear velocity. ALH, mean amplitude of lateral head displacement. BCF, beat cross frequency. STR, straightness. LIN, linearity. NAR, normal apical ridge acrosomes. Values are expressed as means ± SEM. ^a,b^ Values in rows with different superscripts differ significantly at *p* < 0.05.

**Table 2 animals-13-00990-t002:** Coefficients of correlation between semen quality parameters in thawed samples of red deer epididymal sperm. Fluorescence parameters were assessed by fluorescence microscopy.

	TMOT	PMOT	SYBR^+^ /PI^−^	HMP	PNA^−^/PI^−^	YOPRO^−^/PI^−^	DFI
TMOT		0.78 *	0.59	0.73 *	0.67 *	0.68 *	−0.04
PMOT			0.72 *	0.38	0.17	0.15	0.41
SYBR^+^/PI^−^				0.88 **	0.85 **	0.72 *	−0.03
HMP					0.83 **	0.79 **	−0.18
PNA^−^/PI^−^						0.87 **	−0.08
YOPRO^−^/PI^−^							0.28

* Values are significant at *p* < 0.05. ** Values are significant at *p* < 0.01.

**Table 3 animals-13-00990-t003:** Coefficients of correlation between semen-quality parameters in thawed samples of red deer epididymal sperm. Fluorescence parameters were assessed by flow cytometry.

	TMOT	PMOT	SYBR^+^ /PI^−^	HMP	PNA^−^/PI^−^	YOPRO^−^/PI^−^	DFI
TMOT		0.78 *	0.60	−0.23	0.67 *	0.28	0.42
PMOT			0.68 *	−0.19	0.73 *	0.80 *	−0.34
SYBR^+^/PI^−^				−0.12	0.95 ***	0.84 **	0.20
HMP					−0.14	0.04	0.21
PNA^−^/PI^−^						0.89 **	0.12
YOPRO^−^/PI^−^							−0.16

* Values are significant at *p* < 0.05. ** Values are significant at *p* < 0.01.*** Values are significant at *p* < 0.001.

## Data Availability

Not applicable.

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
