# Peer review of "Fluorescence Microscopy and Flow-Cytometry Assessment of Substructures in European Red Deer Epididymal Spermatozoa after Cryopreservation"

_animals, 2023, doi:10.3390/ani13060990_

Round 1

Reviewer 1 Report

This study compares the epididymal sperm quality (before and after cryopreservation) of red deer when was collected within two hours post mortem or after 12 hours of storage at 2-4°C. In addition, two methods for assessing several sperm parameters were compared, fluorescent microscopy (FM) and flow cytometry (FC).  The results indicated the storage of the sperm in the epididymis at for 2-4°C 12 h had little effect on pre-cryopreservation sperm motility parameters, sperm viability, percentage of sperm with high mitochondrial membrane potential, acrosome integrity, and DNA fragmentation. 

The comparison of FM and FC was an interesting addition. It would have been useful to have had FC measures prior to cryopreservation too, especially given the differences between many FM and FC reported for assessment of thawed samples. However, this does not detract from the conclusions drawn from the manuscript. In terms of statistical analyses, was the normality of the data tested prior to analysis using a GLM and Tukey’s test. If so, state which method was used.  

The conclusions could be expanded to include differences between the two assessment methods and how some are more suited to certain parameters. For example, the abstract states that “FM was better suited for evaluating mitochondrial activity”. Such a statement would be worth adding to the conclusions section as well.  The discussion also has some good points regarding the use of FM for the assessment of DNA fragmentation and indicated that FC may be a better method for this parameter, which could be worth  stating in the conclusion. For other parameters, as stated in the conclusions, it seems that both methods can be used to assess various sperm structures. 

Overall, this is a well-written manuscript with sufficiently descriptive methods and results.  The aims and objectives are clearly stated in the introduction and match the conclusions. The introduction and discussion are also well written and supported by appropriate citations. The authors should be commended on putting together a well prepared manuscript. I enjoyed reading it. 

Please see my comments on the attached PDF for more comments (mostly grammatical or formatting comments).

Author Response

Response to Reviewer 1 Comments

This study compares the epididymal sperm quality (before and after cryopreservation) of red deer when was collected within two hours post mortem or after 12 hours of storage at 2-4°C. In addition, two methods for assessing several sperm parameters were compared, fluorescent microscopy (FM) and flow cytometry (FC).  The results indicated the storage of the sperm in the epididymis at for 2-4°C 12 h had little effect on pre-cryopreservation sperm motility parameters, sperm viability, percentage of sperm with high mitochondrial membrane potential, acrosome integrity, and DNA fragmentation. 

Point 1: The comparison of FM and FC was an interesting addition. It would have been useful to have had FC measures prior to cryopreservation too, especially given the differences between many FM and FC reported for assessment of thawed samples. However, this does not detract from the conclusions drawn from the manuscript. In terms of statistical analyses, was the normality of the data tested prior to analysis using a GLM and Tukey’s test. If so, state which method was used.  

Response 1: I agree that it would be a good idea to compare the results of FM and FC assays before cryopreservation. However, such a comparison was not possible for technical reasons because hunting grounds are located more than 500 km away from a laboratory equipped with a flow cytometer. Sperm samples were cryopreserved in the nearest laboratory where only a fluorescence microscope was available, which is why only FM assays could be performed. The assumption of normality was checked using the Shapiro-Wilk test, and the homogeneity of variance was assessed with Levene's test. The data that were not normally distributed were transformed accordingly. The percentage data were subjected to arcsine transformation to obtain normally distributed data before statistical analysis. The significance of differences between group means was determined by Tukey's test. The relevant information was provided in the description of the statistical analysis (lines 248-251).

Point 2: The conclusions could be expanded to include differences between the two assessment methods and how some are more suited to certain parameters. For example, the abstract states that “FM was better suited for evaluating mitochondrial activity”. Such a statement would be worth adding to the conclusions section as well.  The discussion also has some good points regarding the use of FM for the assessment of DNA fragmentation and indicated that FC may be a better method for this parameter, which could be worth  stating in the conclusion. For other parameters, as stated in the conclusions, it seems that both methods can be used to assess various sperm structures.

Response 2: As suggested, the relevant information was included in the abstract and in the conclusions (lines: 42,  434-436).

Overall, this is a well-written manuscript with sufficiently descriptive methods and results.  The aims and objectives are clearly stated in the introduction and match the conclusions. The introduction and discussion are also well written and supported by appropriate citations. The authors should be commended on putting together a well prepared manuscript. I enjoyed reading it. 

Point 3: Please see my comments on the attached PDF for more comments (mostly grammatical or formatting comments).

Response 3: The Reviewer's remarks were addressed during the revision process (changes marked in the manuscript).

Thank you for taking the time to review the manuscript and for the positive feedback.

Reviewer 2 Report

This manuscript is a well written paper evaluating both fluorescence microscopy and flow cytometery evaluation in addition to short term cooled storage prior to cryopreservation. 

I have a few specific comments and suggestions. 

Materials and Methods. 

How was age established and can you provide a range in addition to the mean?

Were the samples paired as in one testis per buck was processed immediately (within 2 hr) vs stored in 2-4 C or were these samples from separate bucks?

page 3 line 134.  Is this spectra adequate for all fluorophores?  Some of the dyes used have other spectra listed for their ideal excitation/emission.

Lines 271 and 389.  Should integral be "intact"

Author Response

Response to Reviewer 2 Comments

This manuscript is a well written paper evaluating both fluorescence microscopy and flow cytometry evaluation in addition to short term cooled storage prior to cryopreservation. 

I have a few specific comments and suggestions. 

Materials and Methods. 

Point 1: How was age established and can you provide a range in addition to the mean?

Response 1: The age of the hunted stags was determined based on a dental analysis. This method has been described in many papers (Mitchell, 1963, Nature 198: 350-351; Lowe, 1967, Journal of Zoology London 152: 137-153). Our study relied on the age assessment method that had been originally developed by Eidmann (1932) and described by Zalewski et al. (2009, Sylwan 153 (4): 240-252). The material for our study was obtained from stags aged 4 to 11 years. The relevant information was added to the manuscript (lines 98-99).

Point 2: Were the samples paired as in one testis per buck was processed immediately (within 2 hr) vs stored in 2-4 °C or were these samples from separate bucks?

Response 2: Samples from different males were used in the study. The material collected from stags (testes with epididymides in the scrotum) after a night hunt was refrigerated (at 2-4 °C) and transported to the laboratory in a portable refrigerator on the following morning. In turn, the material after a morning hunt was transported to the laboratory within several hours. Individual differences could undoubtedly influence the results, but only minor differences in spermatozoa quality were noted in the samples collected several hours after the hunt and the samples refrigerated for 12 hours, which suggests that this factor had a negligent impact on the results.

Point 3: page 3 line 134.  Is this spectra adequate for all fluorophores?  Some of the dyes used have other spectra listed for their ideal excitation/emission.

Response 3: I agree, and the methodology section was revised accordingly (lines 137-138). Fluorochromes differ in excitation and emission spectra, which is why not every fluorochrome can be excited by every laser. The applied fluorochromes had the following excitation/emission wavelengths (nm): Sybr-14 – 488/516; PI–535/617; orange acridine – 500⁄526 (DNA), 460/650 (RNA); JC-1 – 514⁄529, 590; YO-PRO-1/Iodide – 491/509; lectin PNA from Arachis hypogaea (peanut), Alexa Fluor® 488 conjugate – 495/519. The fluorescence microscope was equipped with an appropriate set of filters (including blue and green); therefore, various fluorochromes could be used.

Point 4: Lines 271 and 389.  Should integral be "intact"

Response 4: The relevant correction was made (lines 279 and 397).

Thank you for taking the time to review the manuscript and the comments.
